# AGENT-G: AN AGENTIC FRAMEWORK FOR GRAPH RETRIEVAL AUGMENTED GENERATION

## ABSTRACT

Given two knowledge sources, one containing unstructured documents and the other comprising structured graph knowledge bases, how can we effectively retrieve the relevant information to answer user questions? While Retrieval-Augmented Generation (RAG) retrieves documents to assist the large language model (LLM) in question answering, Graph RAG (GRAG) uses graph knowledge bases as an additional knowledge source. However, there are many questions that require information from both sources, which complicates the scenario and makes hybrid retrieval essential. The goal is to effectively leverage both sources to provide better answers to the questions. Therefore, we propose AGENT-G, a unified framework for GRAG, composed of an agent, a retriever bank, and a critic module. AGENT-G has the following advantages: 1) Agentic, it automatically improves the agent's action with self-reflection, 2) Adaptive, it solves questions that require hybrid knowledge source with a single unified framework, 3) Interpretable, it justifies decision making and reduces hallucinations, and 4) Effective, it adapts to different GRAG settings and outperforms all baselines. The experiments are conducted on two real-world GRAG benchmarks, namely STARK and CRAG. In STARK, AGENT-G shows relative improvements in Hit@1 of *47%* in STARK-MAG and *55%* in STARK-PRIME. In CRAG, AGENT-G increases accuracy by *35%* while reducing hallucination by *11%*, both relatively.

## 1 INTRODUCTION

Retrieval-Augmented Generation (RAG) (Lewis et al., 2020; Guu et al., 2020) enables large language models (LLMs) to have access to the unstructured document database in order to handle unknown facts and reduce hallucinations (Ram et al., 2023; Gao et al., 2023; Béchard & Ayala, 2024). However, RAG often overlooks relationships within the data and lacks a global context. Graph RAG (GRAG) retrieves information from pre-constructed structured graph knowledge bases, providing varying levels of granularity and relationships among text-attributed entities (Edge et al., 2024), which allows the LLMs to better understand their relationships. Although utilizing graph knowledge bases as an additional knowledge source facilitates GRAG in tackling a wider variety of questions, the diversity of data modalities raises two crucial challenges.

First, "*hybrid*" questions that require information from multiple knowledge sources need a tailored solution. While most RAG approaches focus on *textual* questions (Fig. 1 left), which are presumably answerable from text documents, current GRAG approaches (Sun et al., 2024; Jin et al., 2024; Mavromatis & Karypis, 2024) focus on *relational* questions (Fig. 1 middle), assuming that the answers exist in the graphs. As a result, hybrid questions, that require both knowledge sources to be answered correctly (Fig. 1 right), are often neglected. Developing tools to retrieve the necessary information from multiple sources to solve all these questions remains a challenge.

Second, even when the appropriate tools are provided, tool-use methods may find it challenging to operate them correctly on the first attempt. They may (i) perform a retrieval action on an incorrect source (Fig. 1 left), or (ii) retrieve the information with incorrect (action) input (Fig. 1 right). It is therefore essential to provide feedback for them to improve their actions. Despite existing works showing that feedback improves LLM output, their feedback either lacks clear guidance (Yao et al., 2023; Shinn et al., 2023), or requires a fine-tuned model (Paul et al., 2024; Yan et al., 2024). Inferior feedback, especially without careful design, can even mislead the methods when operating the tools.

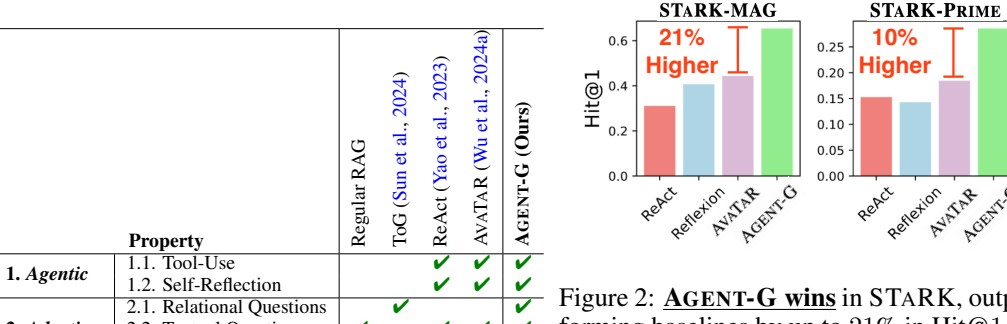

Figure 1: **AGENT-G addresses the challenges in GRAG.** (i) AGENT-G solves textual question (left), relational question (middle), and hybrid question (right) requiring information from both the documents and the graph knowledge bases in GRAG. (ii) In a more complicated GRAG scenario, for a question in which the textual aspect can be confused with the relational aspect, AGENT-G successfully improves its action through self-reflection and answers the question correctly.

Table 1: **AGENT-G matches all properties**, while baselines miss more than one property.

| | Property | Regular RAG | ToG (Sun et al., 2024) | ReAct (Yao et al., 2023) | AVATAR (Wu et al., 2024a) | AGENT-G (Ours) |
|---|---|---|---|---|---|---|
| **1. Agentic** | 1.1. Tool-Use | | | ✔ | ✔ | ✔ |
| | 1.2. Self-Reflection | | | ✔ | ✔ | ✔ |
| **2. Adaptive** | 2.1. Relational Questions | | ✔ | | | ✔ |
| | 2.2. Textual Questions | ✔ | | ✔ | ✔ | ✔ |
| | 2.3. Hybrid Questions | | | | | ✔ |
| **3. Interpretable** | | ? | ✔ | ✔ | ✔ | ✔ |

Figure 2: **AGENT-G wins** in STARK, outperforming baselines by up to 21% in Hit@1.

To address these challenges, we propose AGENT-G, an agentic and unified framework for GRAG. The agent in AGENT-G solves questions that require hybrid knowledge sources with our designed retriever bank, including text and graph retrieval modules. Furthermore, it automatically improves its tool-use action based on feedback from our carefully designed critic module. AGENT-G leverages chain-of-thought (CoT) prompting in each iteration to reduce the chance of answering a question without enough or proper context. We summarize the contributions of AGENT-G as follows:

1. **Agentic**: it automatically improves the tool-use action with self-reflection;
2. **Adaptive**: it solves textual, relational and hybrid questions with a unified framework;
3. **Interpretable**: it justifies the decision making and reduces hallucinations; and
4. **Effective**: it adapts to different GRAG settings and outperforms all the baselines.

In Table 1, compared to the baselines, AGENT-G is the only one that satisfies all properties. We evaluate AGENT-G on two real-world GRAG benchmarks, STARK and CRAG. In Fig. 2, AGENT-G outperforms the second-best baseline by 21% and 10% in STARK-MAG and STARK-PRIME, respectively. In CRAG, AGENT-G increases question answering accuracy by 35% and reduces hallucination by 11%, both relatively.

**Reproducibility:** We will publish the code as soon as we get approval from the legal team.

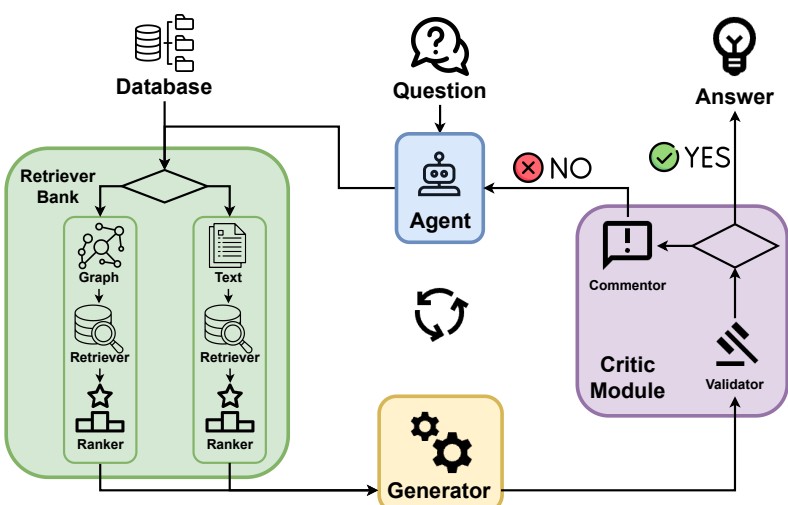

Figure 3: **AGENT-G framework**, which consists of an agent, a retiever bank, a generator, and a critic module. The agent plays the core role in AGENT-G by improving its action to operate the retriever bank based on the feedback from the critic module.

## 2 AGENT-G: AN AGENTIC FRAMEWORK FOR GRAG

Given a question, how can we answer it with the help of information from the most appropriate source? In addition to RAG with access to unstructured documents, GRAG has access to structured graph knowledge bases to assist in question answering. However, how can we leverage information from both sources to answer the question? We first revisit two major challenges for GRAG.

**Challenge 1** (Multi-Sourcing Question). *In GRAG, there are questions that require information from different or multiple sources to be answered.*

As shown in Fig. 1 left, while textual questions require information from documents, relational questions require the information derived from the graph knowledge bases. Some questions even require information from both sources to be answered, which we call "hybrid" questions in Fig. 1 right. Designing a single framework to answer all questions is challenging.

**Challenge 2** (Refinement-Required Question). *Due to the complicated scenario in GRAG, LLMs struggle to answer questions correctly on the first attempt, necessitating refinement of their action.*

The questions in the complicated GRAG scenario involve uncertainty about the relevant knowledge sources and confusing cues for operating the retriever. These cause two common types of retrieval errors that can be made. The first type of error arises when selecting an incorrect retriever. In Fig. 1 middle, retrieving information from documents does not help answer a relational question. The second type of error occurs when the retriever is operated incorrectly. In hybrid questions that contain both relational and textual aspects, LLM can falsely identify the textual aspect as the relational one. In Fig. 1 right, there is an error in retrieving the correct reference by LLM as it confuses the textual aspect as an entity with type "field of study".

Therefore, we propose AGENT-G, an agentic framework for GRAG that consists of a *retriever bank* and a *critic module*, to address Challenge 1 and 2, respectively.

### 2.1 OVERALL FRAMEWORK

The overall framework of AGENT-G is in Fig. 3 and the algorithm is in Algo. 1. The agent plays a core role in AGENT-G by interacting with the retriever bank and the critic module. Given a natural language question $q$, at iteration $t$, the agent determines the action $a_t$ to operate the retriever bank to retrieve reference $\mathcal{X}_t$ from the database $\mathcal{D}$. The generator then produces the output $\hat{y}_t$ with the help of the reference $\mathcal{X}_t$. The critic module decides whether to accept $\hat{y}_t$ as the final answer or reject it. If $\hat{y}_t$ is rejected, it generates feedback $f_t$ for the agent to assist in refining its action at iteration $t+1$.

**Algorithm 1:** AGENT-G Framework

**Input:** Question $q$, Database $\mathcal{D}$, and Maximum Iterations $T$

1   $f_0 = \texttt{""}$;
2   **for** $t = 1, \ldots, T$ **do**
3      $a_t = Agent(q, f_{t-1})$;
4      $\mathcal{X}_t = RetrieverBank(q, a_t, \mathcal{D})$;
5      $\hat{y}_t = Generator(q, \mathcal{X}_t)$;
6      `/* Validator                                    */`
7      **if** $C_{val}(q, \hat{y}_t, \mathcal{X}_t) = True$ **then**
8         Return $\hat{y}_t$;          `// If accepted, return answer`
9      **else**
10        `/* Commentor                                    */`
11        $f_t = C_{com}(q, a_t)$;       `// If rejected, give feedback`
12 Return $\hat{y}_T$;

## 2.2 RETRIEVER BANK

To solve Challenge 1, we propose a retriever bank (in Fig. 3 green), which is composed of multiple retrieval modules. Each retrieval module includes a retriever and a ranker, offering the flexibility to cover a wide range of questions. The retriever first retrieves the most relevant references, and then the ranker gives the ranking of them. At each iteration $t$, top-$K$ references $\mathcal{X}_t$ will be retrieved for answer generation. More specifically, we design two retrieval modules, namely text and graph retrieval modules, to retrieve information from documents and graph knowledge bases, respectively.

The action $a_t$ of an agent includes the selection and input of the retrieval module. The agent first decides the action input by identifying the information of the question $q$, such as the domain $d_t$[1], and extracting the topic entities $\mathcal{E}_t^{(d_t)}$ and useful relations $\mathcal{R}_t^{(d_t)}$ in the question $q$. The agent then makes the selection $s_t$, deciding whether to use a text or a graph retrieval module. Deciding on the input before selection can help optimize the choice of the retrieval module. For example, if there is no entity extracted in this iteration, a text retrieval module can potentially be a better choice. In summary, the action $a_t$ is a set containing $\{d_t, \mathcal{E}_t^{(d_t)}, \mathcal{R}_t^{(d_t)}, s_t\}$.

The text retrieval module retrieves documents using similarity search to the question $q$ such as dense retrieval, which aims to solve textual questions. The graph retrieval module extracts the ego-graph from the graph of the identified domain $d_t$, based on the topic entities $\mathcal{E}_t^{(d_t)}$ and useful relations $\mathcal{R}_t^{(d_t)}$. If there are multiple extracted ego-graphs, it extracts the intersection of them. In addition, we provide two different output formats for the graph retrieval module to handle most types of graph: (i) the reasoning paths by verbalizing the extracted subgraph between the topic entities and the entities in the subgraph, and (ii) the documents associated with the entities in the subgraph.

## 2.3 CRITIC MODULE

In Challenge 2, when faced with a question in the more complicated GRAG scenario, not only the selection of the retrieval module $s_t$ can be incorrect in the first iteration, but also the action input, including the domain $d_t$, topic entities $\mathcal{E}_t^{(d_t)}$, and useful relations $\mathcal{R}_t^{(d_t)}$. To solve this, we propose the critic module (in Fig. 3 purple), which has two parts, a validator LLM $C_{val}$ to validate the correctness of the output $\hat{y}_t$, and a commentor LLM $C_{com}$ to give good feedback $f_t$. Unlike a traditional critic, our essential divide-and-conquer step in the critic module offers two key advantages. First, by breaking a difficult task into two easier ones, we can now leverage pre-train LLMs to solve them while maintaining good performance, instead of fine-tuning a critic LLM with significant resources. Second, this step allows the validator and commentor LLMs to have their own exclusive contexts, preventing including information that is irrelevant to their tasks, which can seriously distract them.

---

[1]There may be multiple domains in graph knowledge bases, such as finance, sports, and movie.

Table 2: Design of retriever bank in AGENT-G for STARK and CRAG.

| Benchmark | Questions | | | Tasks | | Source | Retriever | Reference Type |
|---|---|---|---|---|---|---|---|---|
| | Relational | Textual | Hybrid | Retrieval | Generation | | | |
| STARK | | ✔ | ✔ | ✔ | | Graph
Text | Entities of Ego-Graphs
Dense Retriever | Documents Associated with Entities
Unstructured Documents |
| CRAG | ✔ | ✔ | | ✔ | ✔ | Graph
Text | Entities of Ego-Graphs
Web Search | Reasoning Paths
Web Pages |

Table 3: Design of critic module in AGENT-G for STARK and CRAG.

| Benchmark | Error Source | Error Type | Feedback |
|---|---|---|---|
| STARK | Input | Incorrect Entity/Relation | Entity/relation {name} is incorrect. Please remove or substitute this entity/relation. |
| | | Missing Entity | There is only one entity but there may be more. Please extract one more entity and relation. |
| | | No Intersection | There is no intersection between the entities. Please remove or substitute one entity and relation. |
| | | Incorrect Intersection | There is an intersection between the entities, but the answer is not within it. Please remove or substitute one entity and relation. |
| | Selection | Incorrect Document | The retrieved document is incorrect. The current retrieval module may not be helpful to narrow down the search space. |
| CRAG | Input | Incorrect Question Type | The predicted question type is wrong. Please answer again. Which type is this question? |
| | | Incorrect Question Dynamism | The predicted dynamism of the question is wrong. Please answer again. Which dynamism is this question? |
| | | Incorrect Question Domain | The predicted domain of the question is wrong. Please answer again. Which domain is this question from? |
| | | Incorrect Entity and Relation | The topic entities and useful information extracted from the question are incorrect. Please extract them again. |
| | Selection | Incorrect Retrieval Module | The reference does not contain useful information for solving the question. Should we use knowledge graph as reference source based on newly extracted entity and relation, or use the next batch of text documents as reference source? |

The validator LLM $C_{val}$ aims to identify if the given output $\hat{y}_t$ is correct for the question $q$, which is a binary classification task. To improve accuracy, we provide an additional validation context for the validator LLM. The validation context can be used to verify if the output satisfies certain requirements in the question. For example, if a hybrid question is asking for a document satisfying a relational requirement (e.g., a paper from a specific author), then the context including the reasoning paths is essential for verification (e.g. "{author} →{writes} →{paper}"). In addition, the validation context can be used to determine whether the output is generated based on a reliable reference. In this case, using the retrieved reference $\mathcal{X}_t$ as the context improves the accuracy of the validator.

The commentor LLM $C_{com}$ aims to provide feedback for the agent to improve its action. To effectively guide the agent, we construct *corrective* feedback that the agent can easily understand. In more detail, it points out the error(s) in each action, such as incorrect selection or input of the retrieval module. Unlike natural language feedback, which may cause uncertainty or inconsistency depending on the LLM used, our corrective feedback provides the agent with clear guidance on how to improve its actions. Furthermore, to avoid fine-tuning, it leverages in-context learning (ICL) to provide sophisticated feedback. We collect successful experiences as examples, with each experience $\{a_t, f_t\}$ comprising a pair of action and feedback, which is verified by the ground truth. During inference, the commentor LLM gives high-quality feedback based on multiple pre-collected examples.

## 3 AGENT-G FOR GRAG BENCHMARKS

In this section, we describe how AGENT-G adapts to different real-world GRAG settings. We focus on solving GRAG settings where either the relations have already been constructed among documents, or graphs are provided as an additional knowledge source. Therefore, we select two GRAG benchmarks that satisfy these settings, namely STARK[2] (Wu et al., 2024b) and CRAG[3] (Yang et al., 2024). Table 2 reports the design of the retriever bank for each benchmark, and Table 3 reports the design of the critic module.

---

[2]https://github.com/snap-stanford/stark
[3]https://www.aicrowd.com/challenges/meta-comprehensive-rag-benchmark-kdd-cup-2024

## 3.1 STARK

**Description** We use two datasets from the STARK benchmark[4] (Wu et al., 2024b), STARK-MAG and STARK-PRIME. Each dataset contains a knowledge graph (KG) and unstructured documents associated with some types of entities. The task is to retrieve a set of documents from the database that satisfy the requirements specified in the question. Noting that the majority of questions are hybrid questions, and there are very few textual questions.

**AGENT-G Retriever Bank** In the text retrieval module, we use the vector similarity search (VSS) (Karpukhin et al., 2020) between question and documents in the embedding space as both the retriever and the ranker. In the graph retrieval module, the retriever extracts the ego-graph connected by the useful relations for each topic entity. These documents associated with entities in the ego-graph or the intersection of ego-graphs are then ranked by a VSS ranker.

**AGENT-G Critic Module** Since the task in STARK focuses on the retrieval of hybrid questions, we use the reasoning paths between the topic entities and entities in the extracted ego-graph as the validation context. This helps the validator verify whether the retrieved documents meet the relational requirement in the question. We construct the corrective feedback that points out errors in the action, in order to refine the extracted entity and relation. The successful experiences are verified by the ground truth entities and used for the commentor to perform ICL.

## 3.2 CRAG

**Description** In the CRAG benchmark (Yang et al., 2024), there are KGs from 5 different domains that can be utilized to retrieve useful reference. For each question, a database that includes 50 retrieved web pages and all 5 KGs is given, but the answer is not guaranteed to be on the web pages, KGs, or both. The task is to generate the answer to the question, with or without the help from the retrieved reference. There are textual and relational questions, covering various question types, such as simple, simple with condition, comparison, and multi-hop.

**AGENT-G Retriever Bank** In the text retrieval module, the web search based on the question is used as the retriever, which is done by CRAG ahead of time. The VSS ranker ranks the web pages based on their similarity to the question in the embedding space. In this module, we provide an additional choice for the agent. If the output generated based on the current batch of retrieved web pages is rejected by the validator, the agent can choose to move on to the next batch in the ranking list. In the graph retrieval module, the retriever extracts the ego-graph connected by the useful relations for each topic entity. As there is no document associated with entity, the retriever retrieves the reasoning paths from topic entities to entities in the extracted ego-graphs. The reasoning paths are verbalized as "{topic entity} →{useful relation} →... →{useful relation} →{neighboring entity}", and ranked by VSS.

**AGENT-G Critic Module** The retrieved reference is used as the validation context to check if it is reliable to answer the question. While the ground truth of the retrieval is not available in CRAG, we construct the corrective feedback based on the agent's action and the evaluation. If the graph retrieval module is used and the evaluation is incorrect, then either the retrieval input (extracted entity and relation or the domain) is incorrect, or selecting graph retrieval module is incorrect; if the text retrieval module is used and the evaluation is incorrect, then the information in the current batch of documents is considered as not useful to answer the question.

## 4 EXPERIMENTS

We conduct experiments to answer the following research questions (RQ):

RQ1. **Effectiveness:** How well does AGENT-G perform in real-world GRAG benchmarks?
RQ2. **Interpretability:** How does AGENT-G improve its action based on the feedback?
RQ3. **Ablation Study:** Are all the design choices in AGENT-G necessary?

---

[4]Due to legal issue, one of the datasets in STARK is not included in this article.

Table 4: **AGENT-G wins on STARK.** '*' denotes that only 10% of the testing questions are evaluated due to the high latency and cost of the methods.

| Method | Base Model | STARK-MAG | | | | STARK-PRIME | | | |
|---|---|---|---|---|---|---|---|---|---|
| | | Hit@1 | Hit@5 | Recall@20 | MRR | Hit@1 | Hit@5 | Recall@20 | MRR |
| QAGNN | RoBERTa | 0.1288 | 0.3901 | 0.4697 | 0.2912 | 0.0885 | 0.2123 | 0.2963 | 0.1473 |
| Think-on-Graph* | Sonnet | 0.1316 | 0.1617 | 0.1130 | 0.1418 | 0.0607 | 0.1571 | 0.1307 | 0.1017 |
| Dense Retriever | RoBERTa | 0.1051 | 0.3523 | 0.4211 | 0.2134 | 0.0446 | 0.2185 | 0.3013 | 0.1238 |
| Multi-VSS | ada-002 | 0.2592 | 0.5043 | 0.5080 | 0.3694 | 0.1510 | 0.3356 | 0.3805 | 0.2349 |
| VSS w/ LLM Reranker* | Opus | 0.3654 | 0.5317 | 0.4836 | 0.4415 | 0.1779 | 0.3690 | 0.3557 | 0.2627 |
| ReAct | Opus | 0.3107 | 0.4949 | 0.4703 | 0.3925 | 0.1528 | 0.3195 | 0.3363 | 0.2276 |
| Reflexion | Opus | 0.4071 | 0.5444 | 0.4955 | 0.4706 | 0.1428 | 0.3499 | 0.3852 | 0.2482 |
| AVATAR | Opus | 0.4436 | 0.5966 | 0.5063 | 0.5115 | 0.1844 | 0.3673 | 0.3931 | 0.2673 |
| Text Retrieval Module (VSS) | ada-002 | 0.2908 | 0.4961 | 0.4836 | 0.3862 | 0.1263 | 0.3149 | 0.3600 | 0.2141 |
| Graph Retrieval Module | Sonnet | 0.5028 | 0.5820 | 0.5031 | 0.5373 | 0.2492 | 0.3274 | 0.3366 | 0.2842 |
| **AGENT-G** | Sonnet | **0.6540** | **0.7531** | **0.6570** | **0.6980** | **0.2856** | **0.4138** | **0.4358** | **0.3449** |

**STARK** We use the default evaluation metrics provided by STARK, which are Hit@1, Hit@5, Recall@20 and mean reciprocal rank (MRR), to evaluate the performance of the retrieval task. We compare AGENT-G with various baselines, including recent GRAG methods (QAGNN (Yasunaga et al., 2021) and Think-on-Graph (Sun et al., 2024)); traditional RAG approaches; and self-reflective LLMs (ReAct (Yao et al., 2023), Reflexion (Shinn et al., 2023), and AVATAR (Wu et al., 2024a)). We use "ada-002" as the embedding model for dense retrieval and ranking, as used in the paper.

**CRAG** We use the default evaluation metrics, where an LLM evaluator is used to determine if the predicted answers are accurate, incorrect (hallucination), or missing. An additional three-way scoring $Score_a$ is used, with $1, -1, 0$ for accurate, incorrect, and missing answers. In CRAG, the baselines are required to select the sources and there is no training set. Therefore, we compare AGENT-G with chain-of-though (CoT) LLM, text-only RAG, graph-only RAG, and RAG that concatenates the text and graph references. In addition, we include two self-reflective LLMs (ReAct, Corrective RAG (Yan et al., 2024)), sharing the same retriever bank while using different critics. We use Claude 3 Sonnet as the LLM evaluator, and CoT prompting (Wei et al., 2022) for all generator LLMs. We use "BAAI/bge-m3" (Chen et al., 2024) as the embedding model for dense retrieval and ranking.

In experiments where the base LLM is not specified, we default to using Claude 3 Sonnet as the model. More implementation details are in Appx. A.

## 4.1 EFFECTIVENESS (RQ1)

**STARK** In Table 4, AGENT-G outperforms all baselines significantly in both datasets in STARK, including GRAG and self-reflective baselines. Most baselines are designed to handle either textual or relational questions, and the results have shown that they are not able to handle hybrid questions (Challenge 1). While GRAG approaches perform poorly, our graph retrieval module is the second-best performing method, highlighting the importance of designing an effective graph retrieval module. As there are much fewer textual questions than hybrid questions, the graph retrieval module outperforms the text retrieval module. We also observe that AGENT-G performs much better than the graph retrieval module, indicating that the extracted entity and relation are frequently incorrect in the first iteration (Challenge 2). By tackling Challenge 1 and 2 with a novel design of retriever bank and critic module, AGENT-G has a significant improvement in performance.

**CRAG** In Table 5, AGENT-G outperforms all baselines in CRAG. It is shown that RAGs with a single retrieval module cannot handle both relational and textual questions (Challenge 1). RAG with a concatenated reference also performs poorly due to the lost-in-the-middle phenomenon caused by irrelevant and distracting retrieval results (Shi et al., 2023; Liu et al., 2024). Although our effective retriever bank is provided, self-reflective baselines still find it difficult to answer the question correctly (Challenge 2). Since ReAct heavily relies on the LLM's capability to think and understand natural language feedback, it often fails to improve its action. Without a fine-tuned retrieval evaluator, Corrective RAG cannot identify the usefulness of a reference and thus makes incorrect decisions. This demonstrates the advantages of our critic module with corrective feedback. Furthermore, we show that AGENT-G is robust to the choice of LLM base models. Since it requires no fine-tuning, the base LLM can easily be replaced with a state-of-the-art model to seamlessly improve performance.

Table 5: **AGENT-G wins on CRAG.** All baselines (except CoT LLM) share our retriever bank.

| Method | Llama 3.1 70B | | | | Claude 3 Sonnet | | | |
|---|---|---|---|---|---|---|---|---|
| | Accuracy ↑ | Halluc. ↓ | Missing | Score$_a$ ↑ | Accuracy ↑ | Halluc. ↓ | Missing | Score$_a$ ↑ |
| CoT LLM | 0.4607 | 0.5026 | 0.0367 | -0.0419 | 0.3910 | 0.4052 | 0.2038 | -0.0142 |
| Text-Only RAG | 0.4105 | 0.3685 | 0.2210 | 0.0420 | 0.5034 | 0.3955 | 0.1011 | 0.1079 |
| Graph-Only RAG | 0.4861 | 0.4442 | 0.0697 | 0.0419 | 0.5303 | 0.2974 | 0.1723 | 0.2329 |
| Text & Graph RAG | 0.4120 | 0.3790 | 0.2090 | 0.0330 | 0.5820 | 0.3416 | 0.0764 | 0.2404 |
| ReAct | 0.1745 | **0.2360** | 0.5895 | -0.0615 | 0.4352 | 0.4075 | 0.1573 | 0.0277 |
| Corrective RAG | 0.4509 | 0.4652 | 0.0839 | -0.0143 | 0.4674 | 0.3333 | 0.1993 | 0.1341 |
| **AGENT-G** | **0.5206** | 0.3588 | 0.1206 | **0.1618** | **0.6322** | **0.2959** | 0.0719 | **0.3363** |

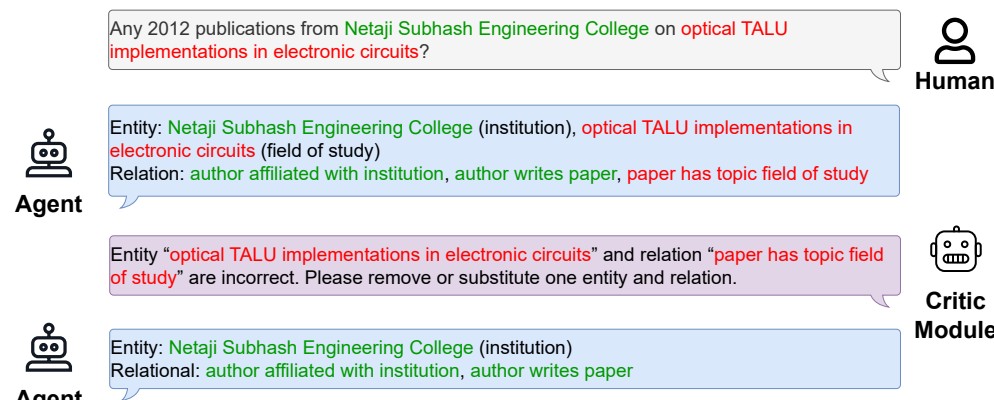

Figure 4: **AGENT-G is interpretable.** In an example from STARK-MAG, the agent understands how to improve action input (entity and relation extraction) based on feedback by the critic module.

## 4.2 INTERPRETABILITY (RQ2)

Fig. 4 shows an example of the interaction between the agent and the critic module in AGENT-G from STARK-MAG. The agent first performs an action that misidentifies "optical TALU implementations in electronic circuits" as a topic entity representing the field of study (relational aspect). While the ego-graph extracted based on this entity has no intersection with the ego-graph extracted based on "Netaji Subhash Engineering College", the critic module identifies that the former entity has a higher chance of being a textual aspect. Thus, it gives the feedback to the agent, and the agent addresses it accordingly. This decision-making process of AGENT-G is similar to chain-of-thought, making it interpretable and easy to understand by the user.

## 4.3 ABLATION STUDY (RQ3)

In this section, we study the effect of different components in AGENT-G, including the design choices in the critic module, and the improvement over iterations by self-reflection.

**Critic Module** We compare AGENT-G variants with validators without validator context, commentors with few or zero shots, and those with oracles. The oracle has access to the ground truth, which gives the optimal judgement on the correctness of the output and the error type of the action, if there is any. In Table 6 and 7, we show that AGENT-G performs the best with all our design choices, approaching the performance of an oracle.

**Self-Reflection** In Figure 5, we demonstrate that with more self-reflection iterations, the performance of AGENT-G improves further. Performance improves significantly when increasing the number of iterations from 1 to 2, where no self-reflection is performed in iteration 1. It is also shown that a few iterations are sufficient, as the improvement diminishes over iterations.

Table 6: **The design choices in AGENT-G are necessary in STARK.** ▮ denotes the settings of AGENT-G, and ▮ denotes the baseline that use ground truth during inference.

| Validator | Commentor | STARK-MAG | | | | STARK-PRIME | | | |
|---|---|---|---|---|---|---|---|---|---|
| | | Hit@1 | Hit@5 | Recall@20 | MRR | Hit@1 | Hit@5 | Recall@20 | MRR |
| w/o Context | ICL | 0.6105 | 0.7073 | 0.6245 | 0.6541 | 0.1946 | 0.2592 | 0.2685 | 0.2251 |
| w/ Context | 5-Shot | 0.6465 | 0.7407 | 0.6458 | 0.6884 | 0.2406 | 0.3006 | 0.3038 | 0.2676 |
| w/ Context | ICL | **0.6540** | **0.7531** | **0.6570** | **0.6980** | **0.2856** | **0.4138** | **0.4358** | **0.3449** |
| Oracle | Oracle | 0.7193 | 0.7824 | 0.6840 | 0.7479 | 0.3606 | 0.4320 | 0.4358 | 0.3932 |

Table 7: **The design choices in AGENT-G are necessary in CRAG.** ▮ denotes the settings of AGENT-G, and ▮ denotes the baseline that use ground truth during inference.

| Validator | Commentor | Accuracy ↑ | Halluc. ↓ | Missing | Score$_a$ ↑ |
|---|---|---|---|---|---|
| w/o Context | ICL | 0.5581 | 0.3461 | 0.0958 | 0.2120 |
| w/ Context | 0-Shot | 0.6277 | 0.3004 | 0.0719 | 0.3273 |
| w/ Context | ICL | **0.6322** | **0.2959** | 0.0719 | **0.3363** |
| Oracle | Oracle | 0.7813 | 0.1640 | 0.0547 | 0.6173 |

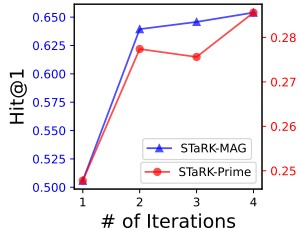

Figure 5: **AGENT-G improves its actions** thanks to the critic module.

## 5 RELATED WORKS

We first introduce the related works of GRAG, including those that have different settings from the one we study, and then the ones that address similar challenges: LLMs with tool-use and self-reflective capabilities. In summary, AGENT-G is the only one that satisfies all properties in Table 1.

**Graph RAG (GRAG)** Different settings have been explored for GRAG (He et al., 2024; Edge et al., 2024; Xu et al., 2024; Peng et al., 2024). While most works are tailored to textual graphs (Wu et al., 2024b) or KGs (Yasunaga et al., 2021; Sun et al., 2024; Jin et al., 2024; Mavromatis & Karypis, 2024), other works (Li et al., 2024; Dong et al., 2024) construct KGs with the given text database.

**LLMs with Tool-Use** Tool-use methods (Trivedi et al., 2022; Yang et al., 2023; Patil et al., 2023; Schick et al., 2023; Gao et al., 2024) seek the tool that generates the best answer. The variability in domains and question types within the graph further complicates the challenge of tool-use. In structured graph data, AVATAR (Wu et al., 2024a) is the most recent work on optimizing the tool-use prompt via contrastive reasoning between positive and negative examples from the training set.

**Self-Reflective LLMs** For complex tasks, LLMs are unlikely to generate the correct output on their first attempt. Self-reflection (Yao et al., 2023; Shinn et al., 2023; Madaan et al., 2023; Paul et al., 2024; Gou et al., 2024; Yan et al., 2024) solves this problem by optimizing the output through an iterative reflection process. A critic is commonly used to give feedback. Previous works use different approaches as critics: pre-trained LLMs (Shinn et al., 2023; Madaan et al., 2023), external tools (Gou et al., 2024), or fine-tuned LLMs (Paul et al., 2024; Asai et al., 2024; Yan et al., 2024).

## 6 CONCLUSIONS

We propose AGENT-G, an agentic framework for Graph Retrieval-Augmented Generation (GRAG). In summary, AGENT-G has following advantages:

1. *Agentic*: it automatically improves the tool-use action with self-reflection;
2. *Adaptive*: it solves textual, relational and hybrid questions with a unified framework;
3. *Interpretable*: it justifies the decision making and reduces hallucinations; and
4. *Effective*: it adapts to different GRAG settings and outperforms all the baselines.

Applied on two GRAG benchmarks, STARK and CRAG, AGENT-G outperforms all baselines. In STARK, AGENT-G achieves an averaged relative improvement $48\%$ in Hit@1; in CRAG, AGENT-G increases the accuracy by $35\%$ while reducing hallucination by $11\%$, both relatively.

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

## A   APPENDIX: REPRODUCIBILITY

### A.1   PROMPTS

#### A.1.1   STARK

The prompt provided to the agent for the first decision making is as follows:

> **You are a helpful, pattern-following assistant. Given the following question, extract the information from the question as requested. Rules: 1. The Relational information must come from the given relational types. 2. Each entity must exactly have one category in the parentheses.**
> **<<<{10 examples for entity and relation extraction}>>>**
>
> **Given the following question, based on the entity type and the relation type, extract the topic entities and useful relations from the question. Entity Type: <<<{entity types}>>>**
> **Relation Type: <<<{relation types}>>>**
> **Question: <<<{question}>>>**
>
> **Documents are required to answer the given question, and the goal is to search the useful documents. Each entity in the knowledge graph is associated with a document. Based on the extracted entities and relations, is knowledge graph or text documents helpful to narrow down the search space? You must answer with either of them with no more than two words.**

The prompt provided to the agent for self-reflection is as follows:

> **The retrieved document is incorrect.**
> **Feedback: <<<{feedback on extracted entity and relation}>>>**
> **Question: <<<{question}>>>**
>
> **The retrieved document is incorrect. Answer again based on newly extracted topic entities and useful relations. Is knowledge graph or text documents helpful to narrow down the search space? You must answer with either of them with no more than two words.**

The prompt provided to the validator LLM is as follows:

> **You are a helpful, pattern-following assistant.**
> **<<<{examples for retrieval validation, 2 for each type of entity}>>>**
>
> **### Question: <<<{question}>>>**
> **### Document: <<<{content of document and reasoning paths}>>>**
> **### Task: Is the document aligned with the requirements of the question? Reply with only yes or no.**

The prompt provided to the commentor LLM is as follows:

> **You are a helpful, pattern-following assistant.**
> **<<<{30 examples of action and feedback pair}>>>**
>
> **Question: <<<{question}>>>**
> **Topic Entities: <<<{extracted entities}>>>**
> **Useful Relations: <<<{extracted relations}>>>**
> **Please point out the wrong entity or relation extracted from the question, if there is any.**

#### A.1.2   CRAG

The prompt provided to the agent for the first decision making is as follows:

**You are a helpful, pattern-following assistant. Given the following question, extract the information from the question as requested. Rules: 1. Each entity must exactly have one category in the parentheses. 2. Strictly follow the examples.**
<<<{examples of entity and relation extraction, 5 for each domain}>>>

**### Question Type: simple, simple_w_condition, set, comparison, aggregation, multi_hop, post_processing, false_premise.**
**### Question:** <<<{question}>>>
**### Task: Which type is this question? Answer must be one of them.**

**### Dynamism: real-time, fast-changing, slow-changing, static.**
**### Question:** <<<{question}>>>
**### Task: Which category of dynamism is this question? Answer with one word and the answer must be one of them.**

**### Domain: music, movie, finance, sports, encyclopedia.**
**### Question:** <<<{question}>>>
**### Task: Which domain is this question from? Answer with one word and the answer must be one of them.**

**Given the following question, based on the entity type and the relation type, extract the topic entities and useful relations from the question.**
**Entity Type:** <<<{entity types}>>>
**Relation Type:** <<<{relation types}>>>
**Question:** <<<{question}>>>

**### Reference Source: knowledge graph, text documents.**
**### Question:** <<<{question}>>>
**### Task: Based on the extracted entity, which reference source is useful to answer the question? You must pick one of them and answer with no more than two words.**

The prompt provided to the agent for reflection is as follows:

**### Question Type: simple, simple_w_condition, set, comparison, aggregation, multi_hop, post_processing, false_premise.**
**### Question:** <<<{question}>>>
**### Task: The predicted question type is wrong. Please answer again. Which type is this question? Answer with one word and the answer must be one of them.**

**### Dynamism: real-time, fast-changing, slow-changing, static.**
**### Question:** <<<{question}>>>
**### Task: The predicted dynamism of the question is wrong. Please answer again. Which dynamism is this question? Answer with one word and the answer must be one of them.**

**### Domain: music, movie, finance, sports, encyclopedia.**
**### Question:** <<<{question}>>>
**### Task: The predicted domain of the question is wrong. Please answer again. Which domain is this question from? Answer with one word and the answer must be one of them.**

**The topic entities and useful information extracted from the question are incorrect. Please extract them again. Given the following question, based on the entity type and the relation type, extract the topic entities and useful relations from the question.**
**Entity Type:** <<<{entity types}>>>
**Relation Type:** <<<{relation types}>>>
**Question:** <<<{question}>>>

**### Reference Source: knowledge graph, text documents.**
**### Question:** <<<{question}>>>
**### Task: The answer is incorrect. The reference does not contain useful information for solving the question. Please answer again, should we use knowledge graph as reference source based on newly extracted entity and relation, or use the next batch of text documents as reference source? You must pick one of them and answer with no more than two words.**

The prompt provided to the generator LLM is as follows:

**You are a helpful, pattern-following assistant.**
**<<<{1 chain-of-though prompt example}>>>**

**### Reference: <<<{reference}>>>**
**### Reference Source: <<<{reference source}>>>**
**### Question: <<<{question}>>>**
**### Query Time: <<<{question time}>>>**
**### Query Type: <<<{question type}>>>**
**### Query Dynamism: <<<{question dynamism}>>>**
**### Query Domain: <<<{question domain}>>>**
**### Task: You are given a Question, References and the time when it was asked in the Pacific Time Zone (PT), referred to as Query Time. The query time is formatted as mm/dd/yyyy, hh:mm:ss PT. The reference may help answer the question. If the question contains a false premise or assumption, answer "invalid question". First, list systematically and in detail all the problems in this problem that need to be solved before we can arrive at the correct answer. Then, solve each sub problem using the answers of previous problems and reach a final solution.**

**What is the final answer?**

The prompt provided to the validator LLM is as follows:

**### Reference: <<<{reference}>>>**
**### Prediction: <<<{output of generator}>>>**
**### Question: <<<{question}>>>**
**### Query Time: <<<{question time}>>>**
**### Task: The prediction is generated based on the reference. Does the prediction answer the question? Answer with one word, yes or no.**

The prompt provided to the commentor LLM is as follows:

**You are a helpful, pattern-following assistant.**
**<<<{5 examples of action and feedback pair}>>>**

**### Reference Source: <<<{reference source}>>>**
**### Question: <<<{question}>>>**
**### Query Time: <<<{question time}>>>**
**### Query Type: <<<{question type}>>>**
**### Query Dynamism: <<<{question dynamism}>>>**
**### Query Domain: <<<{question domain}>>>**
**### Task: Please point out the wrong information about the question (Reference Source, Query Type, Query Dynamism, Query Domain), if there is any. The answer must be one of them.**

The prompt provided to the evaluator LLM is as follows:

**### Question: <<<{question}>>>**
**### True Answer: <<<{ground truth answer}>>>**
**### Predicted Answer: <<<{output of generator}>>>**
**### Task: Based on the question and the true answer, is the predicted answer accurate, incorrect, or missing? The answer must be one of them and is in one word.**

## A.2 EXPERIMENTAL DETAILS

All the experiments are conducted on an AWS EC2 P4 instance with NVIDIA A100 GPUs.

| Domain | Type | Content |
|---|---|---|
| Finance | Entity | company_name, ticker_symbol, market_capitalization, earnings_per_share, price_to_earnings_ratio, datetime |
| | Relation | get_company_ticker, get_ticker_dividends, get_ticker_market_capitalization, get_ticker_earnings_per_share, get_ticker_price_to_earnings_ratio, get_ticker_history_last_year_per_day, get_ticker_history_last_week_per_minute, get_ticker_open_price, get_ticker_close_price, get_ticker_high_price, get_ticker_low_price, get_ticker_volume, get_ticker_financial_information |
| Sports | Entity | nba_team_name, nba_player, soccer_team_name, datetime_day, datetime_month, datetime_year |
| | Relation | get_nba_game_on_date, get_soccer_previous_games_on_date, get_soccer_future_games_on_date, get_nba_team_win_by_year |
| Music | Entity | artist, lifespan, song, release_date, release_country, birth_place, birth_date, grammy_award_count, grammy_year |
| | Relation | grammy_get_best_artist_by_year, grammy_get_award_count_by_artist, grammy_get_award_count_by_song, grammy_get_best_song_by_year, grammy_get_award_date_by_artist, grammy_get_best_album_by_year, get_artist_birth_place, get_artist_birth_date, get_members, get_lifespan, get_song_author, get_song_release_country, get_song_release_date, get_artist_all_works |
| Movie | Entity | actor, movie, release_date, original_title, original_language, revenue, award_category |
| | Relation | act_movie, has_birthday, has_character, has_release_date, has_original_title, has_original_language, has_revenue, has_crew, has_job, has_award_winner, has_award_category |
| Encyclopedia | Entity | encyclopedia_entity |
| | Relation | get_entity_information |

Table 8: Type of entity and relation in the CRAG benchmark.

### A.2.1 DATASETS

**STARK**   We use the testing set from STARK for evaluation, which contains 2665 and 2801 questions for STARK-MAG and STARK-PRIME, respectively. The KG of STARK-MAG is an academic KG, and the one of STARK-PRIME is a precision medicine KG.

**CRAG**   We use the testing set from CRAG for evaluation. There are 1335 textual and relation questions, covering various question types, such as simple, comparison, and multi-hop.

### A.2.2 AGENT-G IMPLEMENTATION

**STARK**   The examples in the prompts are collected from the training set provided by STARK. We use the default entity and relation types provided by STARK. The radius of the extracted ego-graph is no more than two. Four self-reflection iterations have been done. When extracting the entity name from the question, multiple entities in the knowledge base may have exactly the same name. In these cases, we select the entity that has the answer in its one-hop neighborhood for disambiguation, since it is not the focus of our paper. Moreover, these cases rarely happen, where only 3.83% and 0.07% of questions have this issue in STARK-MAG and STARK-PRIME, respectively.

**CRAG**   The examples in the prompts are collected from the validation set provided by CRAG. Since the entity and relation types are not given by CRAG, and the KGs are only accessible with the provided API, we collect them from the questions in the validation set, as shown in Table 8. The radius of the extracted ego-graph is no more than two. Four self-reflection iterations have been done. A batch contains five web pages.

### A.2.3 BASELINE IMPLEMENTATION

**STARK**   We implement Think-on-Graph with their provided code[5]. As running the full experiment takes more than a week, we evaluated it with only 10% of the testing data, as it is done for the LLM reranker in the STARK paper.

**CRAG**   ReAct and Corrective RAG share the same backbone with AGENT-G, while having different critics. ReAct has three actions, namely "search web", "search KG", and "extract entity relation domain", and is given a few examples. The process iterates among action, observation, and thought for four iterations as AGENT-G. While Corrective RAG requires a fine-tuned retrieval evaluator, we implement a version with only a pre-trained LLM. It starts with the text retrieval module and validates if the retrieved reference is correct, ambiguous, or incorrect. If not correct, it uses the graph retrieval module instead. An final answer is generated based on the reference with CoT prompting.

---

[5]https://github.com/GasolSun36/ToG

