# OpenReview forum: "Agent-G: An Agentic Framework for Graph Retrieval Augmented Generation"
_ICLR.cc/2025/Conference — ICLR 2025 Conference Withdrawn Submission_

### Official Review · Reviewer_67mj · 2024-10-22

**Soundness:** 3
**Presentation:** 2
**Contribution:** 2
**Rating:** 3
**Confidence:** 3

**Summary:**

The authors propose a framework for retrieval-augmented generation (RAG) for large language models (LLMs), named AgentG. The system aims to address queries involving both textual and relational problems. It leverages a similarity-based retriever for textual queries and a knowledge graph (KG)-based retriever for relational queries.

**Strengths:**

- **Originality**: To the best of my knowledge, AgentG presented in this paper is an original contribution by the authors.
- **Quality**: The overall quality of the paper is acceptable.
- **Clarity**: The paper is clear in explaining the authors' motivations and methodological design. The structure is simple and easy to follow.
- **Significance**: RAG is a critical research area for LLMs, and constructing retrieval strategies for handling complex queries can have practical implications in real-world applications.

**Weaknesses:**

### Ideas:
- **Textual and Relational Query Types**: The distinction between textual and relational query types is unclear due to the lack of formal definitions. For example, in the query "What is Bob’s mother’s name?", it is not clear which category this query falls under. A more formal definition would help clarify such distinctions.
- **Data Explanation**: The paper doesn't provide a detailed breakdown of the dataset in terms of the proportion of hybrid, textual, and relational queries. Without this, it's hard to assess whether handling hybrid queries is necessary.

### Experiments
- **Model Details**: The paper lacks detailed descriptions of the models used in the experiments. Specifically, it doesn’t clarify which LLMs were used for the agent, validator, and commentator. Additionally, the embedding model for the textual retriever and the specific models for the ego-graph retriever are not mentioned.
- **Baselines**: The paper lacks comparisons with some of the latest methods, such as Microsoft's GraphRAG.
- **STaRK Dataset**: Since the knowledge graph in the STaRK dataset is derived from the documents themselves, why not directly use these documents as the information source?
- **CRAG Dataset**: The CRAG dataset contains only 50 web pages, which seems too small to effectively evaluate RAG performance.

### Analysis
- **Cost**: Handling RAG for knowledge graph queries can be quite expensive due to multiple model invocations per query, leading to high costs. Moreover, the queries addressed in this paper are relatively simple, making the cost-effectiveness even lower.

**Questions:**

- Could you clarify the potential real-world applications where Agent-G could be effectively deployed?
- Assuming Agent-G is the optimal solution for handling hybrid queries, what would be the next steps for future research?

---

### Official Review · Reviewer_enRf · 2024-10-23

**Soundness:** 3
**Presentation:** 2
**Contribution:** 3
**Rating:** 5
**Confidence:** 4

**Summary:**

This paper introduces AGENT-G, an agent framework for Graph RAG scenarios. AGENT-G automatically retrieves information from multiple structured and unstructured knowledge sources based on user queries, allowing the model to generate responses. It uses a critic module to assess the reliability of the generated answers and decide whether to continue retrieving information. On two Graph RAG datasets, AGENT-G outperforms other RAG methods.

**Strengths:**

1. Agent-G can utilize both structured and unstructured knowledge sources simultaneously.

2. Agent-G introduces a Critic Module to evaluate the generated answers, making the responses more robust and interpretable.

**Weaknesses:**

The paper's description of the method is unclear and lacks a formal definition of the entire process, especially regarding how the model determines which knowledge base to retrieve information from. This process is not clearly explained.

1. The novelty of this approach is rather limited; it seems to be a modification of ReAct, allowing the Agent to retrieve information from different types of knowledge sources. By modifying the ReAct's Action function and adding a critic component, a similar functionality can be achieved. Since the paper does not provide code and only offers the corresponding prompt, it is difficult to assess the fundamental differences between this method and ReAct.

2. The inclusion of the Critic Module is a fairly intuitive idea, commonly used in different agent systems. On the other hand, the effectiveness of the Critic Module in this paper heavily depends on the capabilities of the LLM applied to this module.

3. There is no comparison with other Graph RAG methods, such as GraphRAG.

4. The proposed method requires multiple LLM calls for each response, but the paper does not report or compare the number of LLM calls or token consumption for different methods.

**Questions:**

1. Besides being able to retrieve from different knowledge sources and introducing the Critic Module, what are the specific differences between this method and ReAct?

2. How can the reliability of the Critic Module be guarantteed? Especially when the question involves domain knowledge that the LLM Agent hasn't been trained on, can the Critic Module still work effectively?

3. How would the performance of this method change when using smaller LLMs, such as LLAMA3-8B?

4. For each dataset, how many API calls are made per question in this method? What is the average token consumption?

---

### Official Review · Reviewer_ueE4 · 2024-10-27

**Soundness:** 2
**Presentation:** 3
**Contribution:** 1
**Rating:** 3
**Confidence:** 3

**Summary:**

The authors proposed Agent-G, a unified framework for GRAG, which contains an agent, a retriever bank, and a critic module. This framework can automatically improves the agent's action with self-reflection and solves questions which require hybird knowledge source. This framework shows relative improvements in 2 datasets.

**Strengths:**

1.The paper proposed Agent-G, a unified framework for GRAG which can automatically improves the agent's action with self-reflection and solves questions which require hybird knowledge source.

2.The paper is well-organized and presents a clear explanation of the method and the challenges.

3.The experimental results on the two datasets show significant improvements.

**Weaknesses:**

1.The approach presented in the paper appears to be overly simplistic and seems to be a compilation of existing methods without significant innovation. The critic module has no difference with self-reflection metioned in Paragraph 5, while there is no innovation exists in retriever after the agent choose the action.

2.The experiments conducted do not fully explore the performance of the framework with various large language models. It is crucial to consider how different models, especially commonly used ones like GPT-3.5 and GPT-4, perform within this framework.

3.The experiments is insufficient to support conclusions:
- Adaptability: The author claims the framework is adaptive, so it should handle different situations, include traditional graph task like KBQA, GrailQA, etc. The experiments should be conducted on these datasets.
- Interpretability: The paper should provide quantitative metrics to support the interpretability of the work, rather than relying solely on qualitative analysis of a single case. Quantitative indicators(like readability, faithfulness, etc) can help readers better understand the underlying mechanisms and the robustness of the method.

**Questions:**

Recommendations:

- The authors should consider revising the methodology to include more innovative elements that differentiate it from existing methods like self-reflection and RAG technologies.
- The authors should expanding the experimental section to include a wider variety of datasets and models would significantly enhance the paper's contributions to the field. Specifically:
  + Performance of ther large language model like GPT 3.5 and 4
  + Quantitative indicators for interpretability(like readability, faithfulness, etc)
  + Traditional graph task(such as datasets used to evaluate TOG, like KBQA, GrailQA, etc)

---

### Official Review · Reviewer_bVr4 · 2024-10-27

**Soundness:** 2
**Presentation:** 2
**Contribution:** 2
**Rating:** 3
**Confidence:** 2

**Summary:**

This paper addresses Retrieval-Augmented Generation (RAG) in a novel setting, combining both unstructured documents and structured graph-based knowledge. The authors propose AGENT-G, a unified framework for Graph-based Retrieval-Augmented Generation (GRAG) that consists of an agent, a retriever bank, and a critic module.

**Strengths:**

1. The paper explores RAG in a setting that incorporates both unstructured documents and structured graph knowledge bases, which is an novel task setting.
2. The proposed method demonstrates promising performance in preliminary evaluations.

**Weaknesses:**

1. The GRAG task setting requires further clarification. For instance, what specific inputs are expected? How is the knowledge base structured—does it consist of a knowledge graph, or documents with relational data? Figure 1 suggests the latter, but this should be made explicit.
2. The retrieval module includes both a retriever and a ranker, yet the implementation details for these components are not provided. Additionally, the model uses a validator module to confirm if the output is correct for the question, which is a binary classification task. However, the performance of this component is unclear, and justification for its design choice would strengthen the paper.
3. In Table 4, several baseline models appear to use mixed backbone architectures, which may compromise comparability. Furthermore, the table does not clearly explain what these baselines represent (e.g., Cot LLM, Text-only RAG, Text & Graph RAG). It would also be valuable to include other relevant RAG-based baselines, such as [IRCoT](https://aclanthology.org/2023.acl-long.557.pdf), [Adaptive-RAG](https://aclanthology.org/2024.naacl-long.389.pdf), [REPLUG](https://arxiv.org/abs/2301.12652) and [Trace](https://arxiv.org/abs/2406.11460) for a more comprehensive comparison.
4. The writing quality of this paper needs improvement. For example, Line 048 mentions "tool-use methods," which lacks clarity within the context. The role of tools in the introduction is ambiguous and would benefit from justification. Additionally, the experimental section lacks essential details regarding the settings used.

**Questions:**

- Could the authors clearly define the GRAG task and specify the inputs? Specifically, what constitutes the knowledge base—does it take the form of a structured graph, or does it include documents with embedded relations?
- Could the paper provide more implementation details for the retrieval module, particularly the retriever and ranker components? How was the graph retrieval implemented.
- In Line 048, "tool-use methods" is mentioned but seems vague. Could the authors explain what is meant by "tools" in this context and provide justification for their inclusion?

---

### Note · Authors · 2024-11-21

I have read and agree with the venue's withdrawal policy on behalf of myself and my co-authors.